# Efficacy of Lemon Myrtle Essential Oil as a Bio-Fungicide in Inhibiting Citrus Green Mould

**DOI:** 10.3390/plants12213742

**Published:** 2023-10-31

**Authors:** Mohammad M. Rahman, Ronald B. H. Wills, Michael C. Bowyer, Van Q. Vuong, John B. Golding, Timothy Kirkman, Penta Pristijono

**Affiliations:** 1School of Environmental and Life Sciences, University of Newcastle, Ourimbah, NSW 2258, Australia; 2NSW Department of Primary Industries, Ourimbah, NSW 2258, Australia

**Keywords:** lemon myrtle, essential oil, citral, citrus fruit, rind injury, *Penicillium digitatum*, sensory test

## Abstract

The effectiveness of lemon myrtle (LM) (*Backhousia citriodora*) essential oil (EO) was investigated to combat *Penicillium digitatum* by *in vitro* agar diffusion and vapour assay and in artificially infected oranges. The main constituent of LM EO was revealed as citral when analysed in gas chromatography–mass spectrometry. Pure citral was also included in the experiment for comparison. The *in vitro* fungal growth was significantly inhibited by LM EO at 1, 2, 3, 4 and 5 μL per disc while complete growth inhibition by both the pure citral and LM EO occurred at 4 and 5 μL per disc. Inoculated fruits treated by dipping in 1000 μL L^−1^ LM EO solutions for 5, 10, 15, 30 and 120 s showed significantly lower fungal wounds compared to control. While longer dipping times led to some rind injuries, fruits with a 5 and 10 s dip were found free from any injury. The evaluation after dipping and storage confirmed that the fruits maintained the sensory attributes and were not compromised by the incorporation of the essential oil. The results of this study indicate that LM EO can be a promising alternative to synthetic fungicides for preserving the quality of citrus fruits during storage.

## 1. Introduction

The major postharvest disease of citrus fruits is green mould caused by *P. digitatum*, which leads to considerable financial losses for the citrus industry [1]. Chemically synthesised fungicides are routinely used [2], but there are considerable consumer concerns about their use due to potential health and environment effects [3,4]. In addition, the constant and widespread use of chemical fungicides can lead to the development of resistance in target organisms, which can significantly reduce the efficacy of these fungicides [5]. Therefore, the application of natural substances, such as plant essential oils (EOs) can be an alternative approach to control postharvest decay [6]. Plant EOs are primarily composed of terpenoids, which are organic compounds derived from units of isoprene (2-methyl-1,3-butadiene). These compounds play a significant role in the distinct aromas and biological activities of essential oils. Terpenoids have a common structural basis in isoprene units, but their underlying diversity can be accomplished by incorporating a wide range of organic chemical chains, including but not limited to alcohols, aldehydes, ketones, esters, and ethers [7]. The individual constituents of EOs show variable efficacy against postharvest pathogens [6,8,9,10]. There is also interest in EOs with high levels of aldehydes and phenols as such compounds show strong antimicrobial properties [10,11,12].

Lemon myrtle (*Backhousia citriodora*) is an Australian endemic plant, belonging to the family *Myrtaceae*. It is very well known as a medicinal plant and for its lovely lemony fragrance [13,14]. The EO extracted from this plant has been used as a functional ingredient in various products, such as mouthwashes, food flavourings and herbal teas [15,16,17]. Lemon myrtle EO contains a high level of citral (3,7-dimethyl-2-7-octadienal) that consists of the isomeric aldehydes, neral and geranial [18,19]. The antifungal activity of citral in citrus fruits has been reported in several previous studies. For example, Rodov et al. [20] and Ben-Yehoshua and Rodov [21] showed treating fresh citrus fruits with an emulsion consisting of citral effectively decreased the fungal wounds caused by *P. digitatum*. Wuryatmo [22] applied citral as a fumigant in navel oranges stored at low temperatures and found it delayed the onset of green mould. However, this was not effective in preventing green mould when the oranges were stored at ambient conditions. Moreover, multiple research studies have confirmed the occurrence of phytotoxic damage to fruit peels after they were exposed to citral [22,23,24]. Notably, Southwell et al. [25,26] extracted EO from lemon myrtle (LM) leaf and found that this EO not only contains a high level of citral but also has other bioactive compounds, such as myrcene, 6-mcthyl-5-hepten-2-one, linalool, citronellal, iso-neral, iso-geranial, neral and geranial. Wilkinson et al. [27] conducted further tests and found LM EO was effective against 13 bacteria and 8 fungi other than the *Penicillium* genus. Lazar-Baker et al. [28] also reported LM EO was effective in the prevention of mycelium growth and spore germination of *Monilinia fructicola*. 

Although previous studies have linked LM EO with various antimicrobial activities, limited studies have examined the effect of LM EO against *P. digitatum* infection on citrus fruits. Therefore, the objective of the present study was to test the impact of Australian native LM EO on the inhibition of the growth of *P. digitatum* in agar plates and in artificially infected fruits through dipping techniques. The efficacy of inhibition of the LM EO was also compared to that of its main component, citral.

## 2. Materials and Methods

### 2.1. Essential Oils (EO) and Chemicals

Commercial steam-distilled 100% pure lemon myrtle (LM) (*Backhousia citriodora* F. Muell.) EO was purchased from Australian Body (Crafers, South Australia), which sources plant materials grown in broadly dispersed regions of Australia. LM EO was also extracted using a microwave extraction technique in the science laboratory of the University of Newcastle, Ourimbah Campus (NSW), Australia. The LM EO used in all the fruit storage experiments was purchased from Australian Body. Toluene (HPLC Plus grade), citral, limonene, linalool, standard citral, standard limonene, standard linalool and Triton-X100 were purchased from Merck, (Bayswater, VIC, Australia).

### 2.2. Culturing and Preparing Inoculum of P. digitatum 

Cultures of *P. digitatum* (Pers.: Fr.) Sacc. were collected from the Citrus Pathology Laboratory of the New South Wales Department of Primary Industries, Australia. They were maintained and revived on agar medium. For preparing fungal culture for both the *in vitro* experiments and infecting experimental fruits, fungal isolates were sub-cultured on potato dextrose agar (PDA) (Difco Laboratories, Detroit, MI, USA) media in a sterile laminar flow chamber and incubated in a dark chamber at 25 °C for approximately 7 to 10 days. Plastic Petri dishes (Bacto PDS 9014G, NSW, Australia) (85 mm diameter) were used for culture preparation. After 7 days, fungal incubating plates were covered by conidia of *Penicillium* which were collected and diluted with pure water (Reverse Osmosis Plant, GM Autoflow, Newpound, UK) to 10^4^–10^5^ spores mL^−1^. A haemocytometer (Superior, Marienfeld, Germany) and a microscope (Leitz Laborlux S, 100x/1.25, Bruckstr, Burladingen, Germany) were used to determine the spore concentration.

### 2.3. Agar Diffusion Assay

The agar diffusion assay protocol was employed to determine the effectiveness of LM EO in inhibiting the growth of *P. digitatum* according to the methods described by Rodov et al. [29], Javad Safaei Ghomi et al. [30] and CLSI [31], with slight modifications. Briefly, antifungal tests were carried out by employing 100 μL of inoculum suspension containing 10^4^ spores mL^−1^ of fungi spread on a Petri dish (85 mm diameter) containing PDA media. An LM EO sample was pipetted onto a sterile 13 mm antibiotic assay paper disc (Whatman) which was previously placed on the centre of the inoculated agar. As a control, the same amounts of deionised water were pipetted onto an assay paper disc. The Petri dishes were then incubated at 24 °C for 3 days. The antifungal activity of the fungal-growth-free zone around the paper disc was evaluated by measuring the width of the clear zone from the edge of the paper disc to the area of fungal growth and expressed in percentage as percent inhibition. For calculation, the whole width of inhibition including the diameter of the paper disc was measured, then the disc diameter was subtracted to obtain the actual width of inhibition. The width of the Petri dish and the paper disc was 42.5 and 6.5 mm, respectively. Thus, the highest achievable inhibition width was calculated as 36 mm in the case of total growth inhibition. The inhibition was expressed in percentage.

### 2.4. Vapour Assay

In order to evaluate the effects of the oils in the vapour phase, a slightly modified technique of the agar diffusion assay was also used as described by Regnier et al. [32]. Specifically, the sterile 13 mm antibiotic paper disc was stuck at the centre of the lid of the Petri dish and the dish was inverted. This was done to prevent the direct transport of the test compound by diffusion from the paper disc into the agar. The required amounts of test EO doses were pipetted onto the paper discs stuck on the lid. The Petri dishes were sealed with parafilm immediately after the addition of the EOs. The inoculated Petri dishes were kept at 24 °C for 3 days for incubation. The growth of fungal spores was monitored visually by measuring the clear area in the Petri dish and recorded as the inhibition area in mm and expressed as the percent inhibition of the total area.

### 2.5. Laboratory Extraction of EO

EO was extracted from LM leaves using a microwave-assisted extraction system (ETHOS X, Milestone, Sorisole, Italy). The leaves were randomly picked from the naturally growing trees at Ourimbah, NSW, Australia, and authenticated through the herbarium at the University of Newcastle, Australia; they were assigned the voucher number 10638. The extraction process was carried out for 30 min, with the microwave set at 500 watts, and produced 3–3.5 g of oil per 100 g of plant materials. After extraction, EO was kept in a dark-coloured bottle and stored under nitrogen at −20 °C until required.

### 2.6. Gas Chromatography–Mass Spectrometry (GCMS) Analysis

An assessment of both the extracted and commercial EOs was performed using a GCMS (Shimadzu QP 2010SE, Canby, OR, USA) system according to the method described by Rudback et al. [33], with a minor change. In summary, the organic solvent toluene (HPLC Plus grade) was used to prepare a stock solution of oil (35 μL mL^−1^) and stored at −18 °C for further use. The stock solution was diluted to 1000 μL L^−1^ from which an aliquot of one microlitre was delivered to the system. The GCMS system used helium to transport the vaporised solute molecules through the column, maintaining the column flow rate of 3 mL min^−1^ with heating of the injection port at 250 °C, keeping 1 part of the injected sample going to the column and 16 parts going out the split vent. The temperature in the system was programmed at 80 °C with a duration of 3 min and with a gradual increase in temperature of 5 °C per min to an intermediate temperature of 145 °C. Finally, the temperature was increased at 45 °C per min to 275 °C, and this final temperature was held for a duration of 10 min. The temperature for both the ion source and the interface of the equipment was maintained at 250 °C. An SH-Rxi-5Sil MS (Shimadzu) column of 30 m in length was used for the analysis. To identify the oil components, the MS spectra match was validated using the mass spectral database of the National Institute of Standards and Technology (NIST, 2010) installed in the machine as well as with the help of SciFinder. 

### 2.7. Plant Materials

Valencia and Navel orange (*Citrus sinensis* (L.) Osbeck) fruits according to seasonal availability were collected for the investigation. The Navel oranges that were collected from an orchard at Somersby, NSW, Australia, were organic ones. This orchard belonged to the New South Wales Department of Primary Industries (NSW DPI). Again, the Valencia oranges that were collected from a citrus grower at Griffith, NSW, Australia, were commercially matured. All the collected fruits were free from fungicide or waxing treatments after harvest. The selected fruits were sanitised using a sodium hypochlorite solution at a concentration of 10 mL per litre of water. After sanitisation, the fruits were dried at ambient conditions with airflow for approximately 90 min. The air-dried and sanitised fruits were then arbitrarily allotted to the expected number of treatment groups. Each treatment group contained 20 fruits, and there were three to four replicates allocated to each treatment. The same experimental condition was applied to each replicate independently. Experiments were designed to consider variations in the batches of oranges collected at various times of the season.

### 2.8. P. digitatum Inoculation and Fruit Treatment

For inoculation, the flavedo of the fruit was punctured by a steel nail with a diameter of 1 mm and a length of 2 mm dipped in the liquid containing the spores before each piercing. After inoculation, the oranges were incubated at 20 °C and 95–98% RH for 24 h. After inoculation and incubation, the fruit units were then treated individually by dipping in the emulsion containing LM EO or citral for 5, 10, 15, 30 and 120 s. The emulsion was prepared by adding the required volume of citral and LM EO with absolute ethanol at 1 mL L^−1^ as stabiliser and Triton-X100 at 24 mg L^−1^ as surfactant. The emulsion was then diluted to 30 litres of water and blended homogeneously using a blender with a rotation per minute of 550 (Ozito, Guangzhou, China). This ensured that the emulsion was uniformly distributed in the water. The fruits in the control treatment were divided into two groups; one group was treated by dipping only in water, and the other group was treated by dipping in water with the inclusion of ethanol and Triton-X100. 

After treatment, all the fruits were kept on perforated foam net liner placed on plastic trays which were also perforated. All the trays containing the treated fruits were kept at ambient conditions (20–26 °C, 65–75% RH) for one hour to allow the removal of excessive water from the surface of the fruits. Finally, every single replication unit in the treatments was inserted into an open-ended polyethylene bag and kept at 20 °C. Both ends of the bags were folded loosely to provide a balance between maintaining a controlled environment and allowing some interaction with the external atmosphere. The fruits were monitored over a period of 5 to 8 days to determine if they developed any infected lesions on their surfaces. A fruit was considered to have decayed when a soft lesion caused by the fungi exceeded a diameter of 4 mm.

### 2.9. Quality Assessment of Fruits

#### 2.9.1. Rind Injury Assessments

A 1 to 5 scoring scale was used to visually record rind injury on the surface of fruits after treatment and during storage. In this arbitrary scale, 1 = free from any injury, 2 = 1–5% peel injury (minor rind injury with salability), 3 = 6–19% rind damage (reasonable rind injury without salability), 4 = 20–50% rind damage and 5 ≥ =>50% acute rind damage.

#### 2.9.2. Weight Loss

The individual fruit weight loss was measured using a weighing scale (Kern & Sohn GmbH, D-72336, Balingen, Germany) according to the following formula [34]:$$\mathrm{Weight}\;\mathrm{loss}\;\left(\%\right)=\frac{\left(Initial\;weight\;(g)\;-\;Final\;weight\;(g)\right)}{Initial\;weight\;(g)}\times 100$$

### 2.10. Measurement of Fruit Firmness

The individual fruit firmness was determined using a texture analyser (Lloyd Instrument Ltd., Fareham, UK). A compression force (N) in the equatorial zone of the fruit was applied by two flat surfaces of the machine closing together at a speed of 1 mm min^−1^. The force was applied to a depth of 2 mm into the fruit without rupturing the surface. The mean value of the measurements of each fruit was taken from the two sides positioned at a 90° angle according to Cháfer et al. [35] with slight adjustment. 

### 2.11. Respiration Rate

The respiration rate of stored oranges (as evolved CO_2_) was determined as stated by Pristijono et al. [36] with slight modifications. In this procedure, eight fruits were used from each replicate. Firstly, two fruits were placed into each 2L airtight glass container with a rubber stopper in the cap. Then, the jars were kept at 20 °C for five h to accumulate respiratory gases. Finally, a gas sample (1 mL) was collected from the empty space in the container using a syringe. The collected gas sample was transferred to an ICA40 series (International Controlled Atmosphere Ltd., Kent, UK) low-volume gas analysis system. The respiration rate was calculated using the following formula:$$\mathrm{Respiration}\;({\mathrm{mLCO}}_{2}{\mathrm{kg}}^{-1}h^{-1})=\frac{\%\;{\mathrm{CO}}_{2}\;\times \;\mathrm{Vol}.\mathrm{of}\;\mathrm{glass}\;\mathrm{jar}\;(\mathrm{mL})}{\mathrm{Initial}\;\mathrm{fruit}\;\mathrm{weight}\;\left(\mathrm{kg}\right)\times 100\;\times \mathrm{time}\;(h)}$$

### 2.12. Total Soluble Solids 

To determine total soluble solids (TSS), eight fruits from each treatment unit were selected. Juice was extracted using a typical hand extractor and filtered by two sheets of cheesecloth. The strained juice was then placed onto a handy digital refractometer (Atago Co., Ltd., Tokyo, Japan) at 20 °C. By measuring the refractive index of the fruit juice, the TSS was determined and expressed as percentage Brix.

### 2.13. Titratable Acidity

A 5 mL sample of juice was extracted as above. The titratable acidity was determined through a titration process; specifically, a 0.1 M solution of sodium hydroxide was used as the titrant. The titration was carried out until the pH of the juice reached 8.2 using an automatic titrator (Mettler Toledo T50, Greifensee, Switzerland). The data obtained from the titration were expressed as a percentage of citric acid. 

### 2.14. Ethanol

The ethanol content in the oranges was determined by a gas chromatograph–flame ionisation detector (GCFID) with a Carbwax column (Model 580, Gow-Mac, Bethleham, PA, USA) by inserting headspace gas into the system. The temperatures were set as 190 °C, 68 °C and 190 °C for injector, column and FID with gas flow rates at 30, 30 and 300 mL min^−1^ for nitrogen, hydrogen and air, respectively. Firstly, a 10 mL sample of juice was extracted and transferred into a 20 mL glass vial fitted with an aluminium top crimped over the vial’s neck containing a silicone septum. The glass vial containing the juice was then incubated in mild hot water (30 °C) for 10 min. After that, 1 mL of the headspace gas was extracted using a syringe and inserted into the gas chromatograph system. Accordingly, a standard ethanol sample was incubated at the same time, and a 1 mL headspace sample was injected into the system. Ethanol was estimated from the graph generated by the machine and expressed as μL L^−1^.

### 2.15. Ethylene 

For the assessment of ethylene production, a 1 mL headspace gas sample was withdrawn 5 h after sealing the container similar to the previously described method for respiration gas. Ethylene was estimated by inserting the gas sample into a GCFID (Gow-Mac 580, Bridgewater, NJ, USA) fitted with a stainless steel column (2 m × 3.2 mm OD × 2.2 mm ID) packed with Porapak Q (80–100 mesh) (Altech, Sydney, NSW, Australia). The temperature of the detector, column, and injector was set at 110, 90 and 70 °C, respectively. Nitrogen, hydrogen and air were utilised as carrier and combustion gases at a flow rate of 60, 30 and 300 mL min^−1^, respectively. The ethylene production rate was calculated using the following formula:$$\mathrm{Ethylene}\;\mathrm{production}\;(\mu {\mathrm{LC}}_{2}H_{4}{\mathrm{kg}}^{-1}\;h^{-1})=\frac{C_{2}H_{4}\;(\mu LL^{-1})\;\times \;\mathrm{Vol}.\mathrm{of}\;\mathrm{container}\;(L)}{\mathrm{Initial}\;\mathrm{produce}\;\mathrm{weight}\;\left(\mathrm{kg}\right)\;\times \;\mathrm{time}\;(h)}$$

### 2.16. Sensory Evaluation

A discrimination test also known as the triangle test was conducted to determine the difference in any perceptible sensory attributes between Valencia oranges treated by dipping with 1000 μL L^−1^ lemon myrtle (LM) EO and non-treated control oranges according to the method described by Sinkinson [37]. The panellists were provided with assessment questionnaires and were instructed to evaluate the oranges. Each of the panellists was provided three blind-coded slices of oranges; two slices were from one treatment, while one slice was from a different treatment. All three samples were provided in a set order, and panellists were instructed to select the odd sample. Crackers and water were provided to rinse the palate between tasting the orange slices. 

Data interpretation was performed based on the minimum number of correct answers received required for a significance level α ≤ 0.05 according to the one-tailed binomial test [37,38]. The minimum number of correct answers was found in the statistical table [39].

### 2.17. Statistical Analysis

A completely randomised design was utilised incorporating three treatments that were assigned to three to four replicates each. The data obtained were analysed using SPSS version 25 software (Armonk, NY, USA), and the results were represented as the mean ± standard deviation. Multinominal logistic regression analysis was also performed to determine the effects of dipping time in LM EO on the fruits during storage times. In the case of *in vitro* tests, the evaluation was carried out 5 times to confirm the reproducibility. Analysis of variance at a 5% level of significance was used to compare the means, and the least significant difference (LSD) was also calculated using Statistical Analysis System—version 9.4 (SAS Institute, Cary, NC, USA). 

## 3. Results

### 3.1. Composition of LM EO

Table 1 shows the chemical compositions of the extracted and commercial LM EOs. Both the oils analysed contain geranial and neral as the major compounds. Geranial accounts for approximately 50–52% and neral accounts for around 35–39% of the total citral content. Thus, the total citral content in LM EOs ranges from 85% to 91%. Other minor constituents like iso-geranial, benzaldehyde, iso-neral, 6-methyl-5-hepten-2-one, ethylbenzene, and linalool were also revealed in both the extracted and commercial LM EOs. However, the proportion of the minor constituents was very low; each constituent accounted for a range of 0.5 to 2% of the total. 

### 3.2. Efficacy of LM EO and Citral in Agar Diffusion and Vapour Assay

LM EO and citral significantly inhibited fungal growth both in the agar diffusion and vapour assay compared to the non-treated control, as shown in Table 2 and Table 3. The inhibition of fungal growth of the LM EO was significantly higher than that of the control treatments. Moreover, both the LM EO and its principal element citral totally stopped the fungal growth at 4 and 5 μL per disc. 

The germination and growth of *P. digitatum* were effectively inhibited when exposed to LM EO and citral in the agar diffusion assay. With the increase in concentration of the oils from 1 to 5 μL per disc, the efficacy also increased. Furthermore, the vapour assay showed higher efficacy compared to the agar diffusion assay.

### 3.3. Effect of LM EO and Citral on the Fungal Wounds and on the Rind of Oranges

The effects of LM EO and citral against *P. digitatum* in inoculated Navel oranges were examined by dipping 20 fruits for 120 s at 2000–8000 μL L^−1^ LM premixed with Triton-X100 as a surfactant (24 mg L^−1^) and aqueous ethanol (1 mL L^−1^) as a stabiliser. The fruits in the control treatment were dipped in water with only ethanol and Triton-X100 for the same duration of time.

Citral, the principal component of LM, at 1000 μL L^−1^ was also included for comparison. The results showed that LM EO at 2000–8000 μL L^−1^ significantly inhibited the growth of green mould wounds throughout the storage period of 5 days at 20 °C in comparison to the control (Table 4). However, all concentrations of LM as well as citral caused rind injury (Table 5). Increasing the concentrations of LM significantly increased both fungal growth inhibition and rind injury.

This study was extended with inoculated oranges by dipping 20 fruits for 120 s at the lower concentrations of LM from 500–2000 μL L^−1^ and citral at 500 and 1000 μL L^−1^.

The results in Table 6 show that both LM EO and citral were effective in inhibiting fruit wounds caused by *P. digitatum* during storage. After five days of storage, the most effective concentrations of LM EO were found to be between 1000 and 2000 μL L^−1^. The results indicate that there was no significant difference in effectiveness among these LM concentrations in inhibiting green mould growth. Therefore, the optimum efficacy was attained at a concentration of 1000 μL L^−1^. 

Table 7 indicates that when fruits were dipped in LM EO and citral at concentrations ranging from 500 to 2000 μL L^−1^ for a duration of 120 s, it resulted in rind injury. Although the rind injury was present immediately after dipping, it did not increase in severity during the storage period. The severity of rind injury varied depending on the concentration of LM EO and citral. The level of injury was described as “mild” for fruits dipped in 500 μL L^−1^ of LM EO, but it was more severe at concentrations greater than or equal to 1000 μL L^−1^. 

Further examination was conducted with LM EO at 1000 μL L^−1^ with the aim of determining the lowest dipping time that generates an optimal balance between maximising fungal growth inhibition and minimal rind injury.

Inoculated Valencia oranges were examined to determine green mould inhibition after dipping 20 fruits in 1000 μL L^−1^ LM EO for 5, 10, 15 and 30 s. Control fruits were dipped in water with ethanol and Triton-X100. In addition, an extra control water dip without the wetting agent (Triton-X100) and stabiliser (ethanol) was used to confirm that these substances had no impact on mould growth. The results in Figure 1 show firstly that there was no significant difference {χ^2^ (1) = 0.69, *p* = 0.41} between the two control treatments. The results in Figure 1 also show that there is a significant difference {LRT χ^2^ (4) = 56.01, *p* = 0.00} between the inoculated control fruits and inoculated fruits treated with 1000 μL L^−1^ lemon myrtle EO by dipping during the storage times. A 40% rotting of the fruits occurred in the control fruits after 3 days of storage, and it turned to 100% after 5 days. On the contrary, dipping the inoculated oranges in the emulsion of LM EO at the concentration of 1000 μL L^−1^ for a time duration of 5, 10, 15 and 30 s showed a significantly lower rotting percentage compared to control fruits throughout the storage time. On the 3rd and 4th day of storage, on average less than 2 and 20 percent of the inoculated fruits rotted, respectively. Interestingly, the effect of dipping time in 1000 μL L^−1^ lemon myrtle EO on the inoculated oranges is equivalent according to the likelihood ratio test in the multinominal logistic regression model {LRT χ^2^ (3) = 0.00, *p* = 1.00}.

The dataset presented in Table 8 shows that rind injury was observed on a few fruits dipped in LM EO, while no such injury was observed on control fruits. The extent of rind injury seems to be related to the duration of the dip in the LM EO. After 15 and 30 s of dipping, there were some minor rind injuries with mean scores ranging from 1.2 to 1.3, as indicated in Table 8. However, there was no observed injury on fruits treated in LM EO emulsion for 5 and 10 s dip times.

### 3.4. Quality Assessment Study of LM EO-Treated Valencia Oranges 

Valencia oranges were treated with 1000 μL L^−1^ LM EO and citral solution by the dipping method for 30 s, followed by storage at 20 °C and 65% RH for four weeks. The impacts of dipping on physiological weight loss, firmness, rate of respiration, total soluble solids (TSS), titratable acidity (TA), ethanol and ethylene production are listed in Table 9 and Table 10. The findings presented in these tables confirmed that there was no significant influence of LM EO or citral on weight loss, firmness, respiration rate, TSS, TA, ethanol and ethylene production compared to the control fruits.

A discrimination triangle test was conducted to determine if a perceptible sensory difference existed between oranges dipped in 1000 μL L^−1^ LM EO premixed with TritonX-100 (24 mgL^−1^) and aqueous ethanol (1 mL L^−1^) for 30 s and control fruits dipped in water.

Out of the 21 panellists, only 5 were able to correctly identify which samples were treated with LM EO and untreated fruits. This number is less than the 12 required for a significant difference between treatments at *p* = 0.05 [35]. Therefore, it was concluded that treating with LM EO did not result in a negative impact on the taste of the oranges. This suggests that the flavour or palatability of the oranges remained unaffected by the inclusion of LM EO.

## 4. Discussion

Most of the components of the LM EO in the current study are consistent with the components reported by Southwell et al. [25], Kurekci et al. [40] and Buchbauer et al. [41]. The LM EO, which is predominantly rich in citral, effectively prevented the germination and growth of *P. digitatum* in the agar diffusion assay, and with the increase in concentration of the oils from 1 to 5 μL per disc, the efficacy also increased. The complete inhibition of the fungal growth at 4 and 5 μL per disc implies strong biological activity of the oil on the pathogen. The antifungal efficacy of LM EO and citral in this study aligns with the previous findings by Sultanbawa [39], Wilkinson et al. [27] and Lazar-Baker et al. [28] who reported LM EO as an effective antimicrobial agent in an *in vitro* test against 13 bacteria, 1 yeast and 8 fungi other than the *Penicillium* spp. There was greater antifungal efficacy of citral and LM EO in the vapour assay than in the agar diffusion assay which was considered due to the vapour providing better penetration into the agar media than the diffusion assay based on the fact of the oil polarity. Essential oils are composed of non-polar molecules. The diffusion of non-polar molecules through agar is more difficult, whereas vapour has direct contact with the fungus that could inhibit the growth directly. This is supported by the findings of Rodov et al. [29] who reported the antifungal efficacy of citral against *P. digitatum* in vapour assay as almost double in comparison to agar diffusion assay.

The inhibitory action of citral against *P. digitatum* could be explained based on its molecular structure. Citral is an aliphatic aldehyde that contains double bonds conjugated to its carbonyl group. Kurita et al. [42] reported that the fungal inhibition activity of a substance like citral is associated with the energy of its lowest empty molecular orbital. This is because citral has the capability to form a charge transfer complex, particularly with electron donors like tryptophan. When citral forms a charge transfer complex with an electron donor, it involves the transfer of electrons, which can result in the formation of a complex or a new chemical entity. Fungal cells contain electron donors, and citral can form charge transfer complexes with these electron donors present in the fungal cells. The formation of charge transfer complexes involving citral and electron donors in fungal cells is associated with the inhibition of fungal growth. This interaction disrupts or inhibits the growth and activity of the fungus [40,42,43]. The *in vitro* findings of the current study indicate that LM EO’s high citral content has the potential application in controlling fungal diseases in oranges, i.e., *in vivo* efficacy.

Navel oranges dipped in different concentrations of LM EO and citral for 120 s effectively controlled the fungal wounds but also generated rind injury, implying the complexity of balancing between pathogen control and fruit rind injury. So, the use of LM EO at 1000 μL L^−1^ as an attained optimal concentration to determine the lowest dipping time was challenging for this study. Screening LM EO for lower dipping times showed a significant difference in fungal inhibition due to dipping time with the magnitude of reduction in inhibition efficacy being 120 s > 30 s ≈ 15 s ≈ 10 s ≈ 5 s. There was no rind injury on fruits dipped for 5 and 10 s, while minor rind injury was observed on fruits dipped for 15 and 30 s. Comparatively, a higher extent of rind injury was detected on fruits dipped for 120 s. LM EO was selected for evaluation due to its high level of citral, but the application of 1000 μL L^−1^ LM EO on inoculated Navel oranges was significantly more effective in inhibiting mould growth than 100% citral after 5 days of storage. However, the higher impact of inhibiting mould growth of LM EO does not seem to be attributed to variations in geranial and neral proportions in citral and LM EO. It appears that the greater inhibitory effect of LM EO on fungal wastage is likely due to the presence of minor compounds within LM EO that possess stronger antifungal activity compared to the citral constituents (geranial and neral) or their combinations. These minor compounds in LM EO may have more potent antifungal properties, contributing to its effectiveness in preventing fungal wastage on the fruits.

The obvious advantage of using EOs extracted from plants over synthetic fungicides is that they can be considered to be derived from “natural” sources. This would have marketing value for consumers who are suspicious of synthetic chemicals being added to foods, and these EOs should also be acceptable for use on organic citrus fruits. Essential oils in this study are hypothesised to inhibit the growth of *P*. *digitatum* by deforming the mitochondrial morphology, being involved in arresting the respiratory metabolism leading to a decrease in the activities of tricarboxylic acid cycle (TCA)-related enzymes and changing the metabolic abilities of TCA. 

The results obtained in this study, where dipping in LM EO gave good inhibition of *P. digitatum* in Navel and Valencia oranges, indicate that LM EO is worthy of further investigation for its ability to control other mould wounds of citrus fruits, and maybe other fungi on other fruits. While the dipping of oranges in the LM EO did not have any adverse effect on the internal quality and sensory attributes, the potential for the generation of rind injury is indeed a significant concern when considering the use of LM EO for fresh fruit marketing. Rind injury can affect the appearance of oranges, which can have a direct impact on their marketability. The rind injury would seem to be due to the action of citral. The disruption of the cell membrane structure is a common mechanism by which citral exerts its antimicrobial effects. It can affect the integrity and permeability of microbial cell membranes, leading to cell damage and ultimately inhibiting their growth [23,44]. However, this action on cell membranes can also potentially lead to rind injuries in fruits. Striking the right balance between inhibiting pathogen growth and minimising harm to the fruit’s outer structure is a challenge when using EOs for postharvest treatment. The potential phytotoxic injury of citral on the fruit rind is also confirmed by the findings of Knight [24] and Wuryatmo [45] who reported rind injury was more severe on fruits in direct contact with citral. In the current study, fruits treated by dipping in 100% citral exhibited more rind injury than those treated by dipping in LM EO, which contains only 85% citral, suggesting the overall composition of LM EO, including the presence of other compounds, plays a significant role in mitigating rind injury compared to pure citral. 

Shorter dipping times result in less citral being absorbed or accumulating on the fruit’s surface. As a result, the concentration of citral applied to the rind is lower during the shorter dipping durations, reducing the potential for rind injury. For commercial dipping practices, immersing fruits for 30 s appears to be the lowest possible throughput time to avoid the possibility of rind injury. A 30 s dip in LM EO was sufficient to inhibit the development of green mould; however, there was still some level of rind injury observed on certain fruits. This trade-off suggests that while a shorter dip duration is effective in controlling mould, it might not eliminate the risk of rind injuries completely. Dipping in LM EO for 5 and 10 s did not cause rind injury to the fruits, but these shorter dipping times were less effective in inhibiting mould growth when compared to a 120 s dip. To overcome the challenge of rind injury and maintain longer exposure times, it would seem to be worthwhile to explore various application methods of LM EO such as a combination with edible films, coatings and/or nanoencapsulation. Rodov et al. [29] observed that the combination of citral and ethanol at a concentration of 25% (*v*/*v*) was a successful treatment for controlling the decay of “Eureka” lemons caused by the *P. digitatum* fungus. Notably, the application of this combination did not result in visible damage to the rind of the lemons. 

## 5. Conclusions

Both extracted and commercial LM EOs contain high levels of citral with a content of approximately 88%. Other constituents were also found in these LM EOs, though their extent was very low, within a range of 0.5 to 2% of total constituents. LM EO was effective in inhibiting mould growth in the *in vitro* tests and in oranges. In the *in vitro* tests. the crude oil was used without any dilution in the present investigation. So, minimum inhibitory concentration (MIC) and minimum bactericidal concentration (MBC) by microdilution were not determined. The level of mould inhibition of LM EO was even greater than that of citral, the major and well-known antimicrobial compound. This indicates that LM EO might have some as yet unknown minor components that might exhibit potent antimicrobial activity. Therefore, future studies are suggested to examine the MIC and MBC of LM EO and the individual and synergistic effects of other minor components in LM EO on the prevention and treatment of bacterial or other fungal growth in oranges and other fresh produce.

## Figures and Tables

**Figure 1 plants-12-03742-f001:**
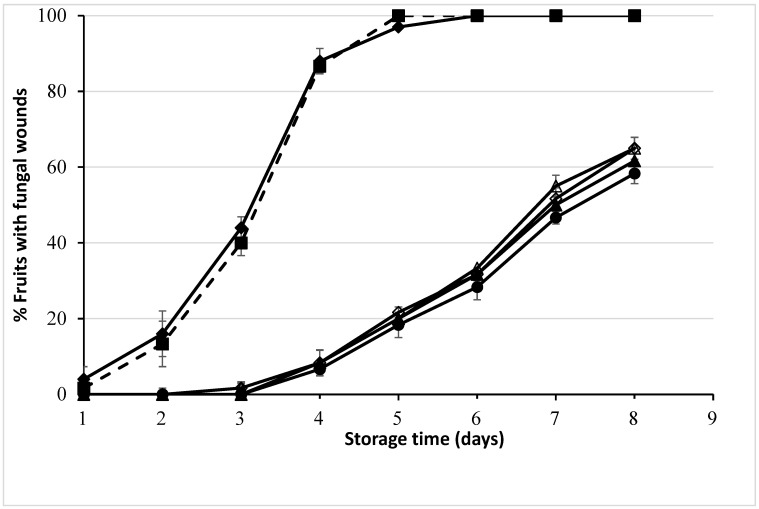
Fungal wounds in Valencia oranges during storage at 20 °C after the oranges were dipped in a solution containing 1000 μL L^−1^ lemon myrtle essential oil (LM EO) for varying durations. Main control (water only) 30 s (♦); additional control (water + Triton-X100 + ethanol) 30 s (■); LM EO dip 5 s (∆), 10 s (◊), 15 s (▲), 30 s (●). Mean of 6 units of 20 fruit (2 groups × 3 repeats). The effect of dipping time in LM EO is equivalent according to the likelihood ratio test in the multinominal logistic regression model {LRT χ^2^ (3) = 0.00, *p* = 1.00}. The curve is fitted to the logistic regression basic equation, log(*p*/(1 − *p*)) = intercept + slope × Days, where p is the proportion of rotten fruits.

**Table 1 plants-12-03742-t001:** Chemical constituents of lemon myrtle essential oil (LM EO).

Retention Time (min)	Laboratory Extracted LM EO		Commercial LM EO	
	Percentage (%) ^a^	Component	Percentage (%) ^a^	Component
3.06	1.0	Ethylbenzene	0.18	Ethylbenzene
4.90	0.6	6-methyl-5-hepten-2-one	0.31	6-methyl-5-hepten-2-one
5.03	0.6	β-Myrcene	0.25	β-Myrcene
7.70	0.5	Linalool	0.35	Linalool
9.38	1.4	Iso-neral	1.75	Iso-neral
11.60	35.3	Neral	38.50	Neral
11.90	2.0	Iso-geranial	0.31	Iso-geranial
12.43	49.5	Geranial	51.60	Geranial

^a^ Relative peak area percent of total peak area.

**Table 2 plants-12-03742-t002:** Antifungal effect of lemon myrtle (LM) essential oil (EO) and citral on the radial growth of *P. digitatum* using the agar diffusion assay after 3 days at 24 °C.

Treatments	Growth Inhibition (Arcsine %)				
	1	2	3	4	5 μL Per Disc
Control	1 ± 2 × 10^−16^	1 ± 2 × 10^−16^	1 ± 2 × 10^−16^	1 ± 2 × 10^−16^	1 ± 2 × 10^−16^
LM EO	27 ± 2	51 ± 2	67 ± 3	89 ± 3 × 10^−14^ *	89 ± 3 × 10^−14^ *
Citral	34 ± 3	61 ± 3	69 ± 2	89 ± 3 × 10^−16^ *	89 ± 3 × 10^−16^ *
*LSD*	*1.4*	*1.8*	*1.9*	*1.3 × 10^−10^*	*1.3 × 10^−10^*

The data are the arcsine transform value of the percent of plate area covered without spores. Values are the means for 15 replicates ± standard deviation (5 groups × 3 repeats). * Full or 100% growth inhibition.

**Table 3 plants-12-03742-t003:** Antifungal effect of lemon myrtle (LM) essential oil (EO) and citral on the radial growth of *P. digitatum* using the vapour assay after 3 days at 24 °C.

Treatments	Growth Inhibition (Arcsine %)				
	1	2	3	4	5 μL Per Disc
Control	1 ± 2 × 10^−16^	1 ± 2 × 10^−16^	1 ± 2 × 10^−16^	1 ± 2 × 10^−16^	1 ± 2 × 10^−16^
LM EO	37 ± 2	60 ± 4	67 ± 3	89 ± 5 *	893 × 10^−14^ *
Citral	45 ± 1	62 ± 2	78 ± 8	89 ± 3 *	89 ± 3 × 10^−14^ *
*LSD*	*1.0*	*1.6*	*3.2*	*4.0 × 10^−2^*	*3.0 × 10^−6^*

The data are the arcsine transform value of the percent of plate area covered without spores. Values are the means for 15 replicates ± standard deviation (5 groups × 3 repeats). * Full or 100% growth inhibition.

**Table 4 plants-12-03742-t004:** Fungal wounds of Navel oranges treated with a solution of lemon myrtle (LM) essential oil (EO) and citral by dipping for 120 s following storing at 20 °C.

Treatments	Wounds (Lesion Diameter, mm)		
	3	4	5 Days
Control	14 ± 4	29 ± 16	44 ± 19
LM EO (μL L^−1^)			
2000	0 ± 0	4 ± 8	9 ± 14
4000	0 ± 0	2 ± 4	5 ± 9
6000	0 ± 0	1 ± 3	4 ± 7
8000	0 ± 0	1 ± 4	3 ± 9
Citral (μL L^−1^)			
1000	0 ± 0	4 ± 11	8 ± 19
*LSD*	*1.8*	*3.2*	*5.2*

Mean values of 3 units of 20 fruits ± standard deviation (1 group × 3 repeats).

**Table 5 plants-12-03742-t005:** The extent of peel injuries on navel oranges dipped for 120 s in emulsions containing lemon myrtle (LM) essential oil (EO) and citral following storing at 20 °C.

Treatments	Peel Injury Score				
	1	2	3	4	5 Days
Control	1.0 ± 0.0	1.0 ± 0.0	1.0 ± 0.0	1.0 ± 0.0	1.0 ± 0.0
LM EO (μL L^−1^)					
2000	1.9 ± 0.2	1.9 ± 0.2	2.1 ± 0.2	2.7 ± 0.1	2.7 ± 0.2
4000	2.2 ± 0.2	2.2 ± 0.2	2.4 ± 0.2	3.1 ± 0.1	3.1 ± 0.1
6000	2.5 ± 0.1	2.5 ± 0.1	2.7 ± 0.1	3.2 ± 0.2	3.2 ± 0.1
8000	2.8 ± 0.2	2.8 ± 0.2	3.0 ± 0.2	3.4 ± 0.1	3.4 ± 0.2
Citral (μL L^−1^)					
1000	1.4 ± 0.2	1.4 ± 0.1	1.6 ± 0.2	1.7 ± 0.2	1.7 ± 0.1
*LSD*	*0.2*	*0.2*	*0.2*	*0.2*	*0.2*

Values are the mean of 3 units of 20 fruits ± standard deviation (1 group × 3 repeats). Scoring scale: 1 = free from any injury, 2 = 1–5% peel injury (minor rind injury with salability), 3 = 6–19% rind damage (reasonable rind injury without salability), 4 = 20–50% rind damage and 5 ≥ 50% acute rind damage.

**Table 6 plants-12-03742-t006:** Fungal wounds of Navel oranges treated with a solution of lemon myrtle (LM) essential oil (EO) and citral by dipping for 120 s following storing at 20 °C.

Treatments	Wounds (Lesion Diameter, mm)		
	3	4	5 Days
Control	20 ± 14	37 ± 18	57 ± 24
LM EO (μL L^−1^)			
500	2 ± 6	14 ± 17	27 ± 27
1000	1 ± 5	5 ± 11	13 ± 25
1250	1 ± 5	5 ± 12	13 ± 19
1500	1 ± 4	3 ± 9	12 ± 18
2000	1 ± 3	4 ± 9	12 ± 16
Citral (μL L^−1^)			
500	3 ± 7	10 ± 14	26 ± 26
1000	2 ± 6	7 ± 12	19 ± 20
*LSD*	*1.9*	*3.5*	*5.4*

Mean of 6 units of 20 fruits ± standard deviation (2 groups × 3 repeats).

**Table 7 plants-12-03742-t007:** The extent of rind injuries on Navel oranges dipped for 120 s in emulsions containing lemon myrtle (LM) essential oil (EO) and citral following storing at 20 °C.

Treatments	Peel Injury Score				
	1	2	3	4	5 Days
Control	1.0 ± 0.0	1.0 ± 0.0	1.0 ± 0.0	1.0 ± 0.0	1.0 ± 0.0
LM EO (μL L^−1^)					
500	1.3 ± 0.5	1.3 ± 0.5	1.3 ± 0.5	1.3 ± 0.5	1.4 ± 0.6
1000	3.2 ± 0.8	3.2 ± 0.8	3.2 ± 0.8	3.2 ± 0.8	3.3 ± 0.7
1250	3.2 ± 0.9	3.2 ± 0.9	3.2 ± 0.9	3.2 ± 0.9	3.4 ± 0.6
1500	3.2 ± 0.6	3.2 ± 0.6	3.2 ± 0.6	3.3 ± 0.6	3.5 ± 0.6
2000	3.3 ± 0.7	3.3 ± 0.8	3.3 ± 0.8	3.3 ± 0.8	3.6 ± 0.6
Citral (μL L^−1^)					
500	1.3 ± 0.5	1.4 ± 0.6	1.4 ± 0.5	1.4 ± 0.6	1.5 ± 0.6
1000	3.4 ± 0.6	3.4 ± 0.6	3.5 ± 0.7	3.5 ± 0.7	3.6 ± 0.7
*LSD*	*0.2*	*0.2*	*0.2*	*0.2*	*0.2*

Mean of 6 units of 20 fruits ± standard deviation (2 groups × 3 repeats). Scoring scale: 1 = free from any injury, 2 = 1–5% peel injury (minor rind injury with salability), 3 = 6–19% rind damage (reasonable rind injury without salability), 4 = 20–50% rind damage and 5 ≥ 50% acute rind damage.

**Table 8 plants-12-03742-t008:** The extent of rind injuries on Navel oranges dipped in lemon myrtle (LM) essential oil (EO) at 1000 μL L^−1^ for different times following storing at 20 °C.

Dip Time (s)	Peel Injury Score				
	1	2	3	4	5 Days
(Control 1) 30	1.0 ± 0.0	1.0 ± 0.0	1.0 ± 0.0	1.0 ± 0.0	1.0 ± 0.0
(Control 2) 30	1.0 ± 0.0	1.0 ± 0.0	1.0 ± 0.0	1.0 ± 0.0	1.0 ± 0.0
5	1.0 ± 0.0	1.0 ± 0.0	1.0 ± 0.0	1.0 ± 0.0	1.0 ± 0.0
10	1.0 ± 0.0	1.0 ± 0.0	1.0 ± 0.0	1.0 ± 0.0	1.0 ± 0.0
15	1.1 ± 0.4	1.2 ± 0.4	1.2 ± 0.4	1.2 ± 0.4	1.2 ± 0.4
30	1.2 ± 0.4	1.3 ± 0.4	1.3 ± 0.4	1.3 ± 0.5	1.3 ± 0.5
*LSD*	*0.1*	*0.1*	*0.1*	*0.1*	*0.1*

Mean of 6 units of 20 fruits ± standard deviation (2 groups × 3 repeats). Main control (water only dip) 30 s; additional control (water + wetting agent + ethanol dip) 30 s. Scoring scale: 1 = free from any injury, 2 = 1–5% peel injury (minor rind injury with salability), 3 = 6–19% rind damage (reasonable rind injury without salability), 4 = 20–50% rind damage and 5 ≥ 50% acute rind damage.

**Table 9 plants-12-03742-t009:** Effects of dipping oranges in an emulsion of lemon myrtle (LM) essential oil (EO) and citral at a concentration of 1000 μL L^−1^ on weight loss, firmness and respiration rate following storing at 20 °C.

Quality Parameters/Dipping Treatments (1000 μL L^−1^)	Weeks				
	1	2	3	4	Mean
Weight loss (%)					
Time—0	0				
Control	2.7 ± 11.6	3.9 ± 12.9	4.8 ± 13.5	5.4 ± 12.6	4.2 ± 12.7
LM EO	2.4 ± 14.7	3.7 ± 12.9	4.8 ± 14.0	5.4 ± 14.0	4.1 ± 13.9
Citral	2.4 ± 13.9	3.6 ± 12.5	4.8 ± 13.5	5.6 ± 14.4	4.1 ± 13.6
*LSD*					*2.1*
Firmness (N)					
Time—0	33.5				
Control	27.9 ± 3.3	24.5 ± 2.6	23.4 ± 3.7	22.8 ± 3.1	27.9 ± 3.9
LM EO	26.2 ± 1.9	25.7 ± 3.7	23.9 ± 2.1	22.5 ± 1.9	26.2 ± 3.4
Citral	25.9 ± 2.7	25.1 ± 2.9	23.4 ± 2.3	22.5 ± 2.7	25.9 ± 3.5
*LSD*					*0.5*
Respiration (mLCO_2_kg^−1^h^−1^)					
Time—0	8.1				
Control	8.6 ± 1.4	10.5 ± 2.0	11.3 ± 0.7	12.6 ± 1.5	10.8 ± 2.0
LM EO	8.6 ± 1.1	10.3 ± 1.8	11.4 ± 0.5	12.5 ± 2.2	10.7 ± 2.1
Citral	8.8 ± 0.8	10.5 ± 1.2	11.4 ± 1.8	12.5 ± 1.5	10.8 ± 1.8
*LSD*					*0.7*

Mean of 4 units of 20 fruits ± standard deviation (1 group × 4 repeats).

**Table 10 plants-12-03742-t010:** Effects of dipping oranges in an emulsion of lemon myrtle (LM) essential oil (EO) and citral at a concentration of 1000 μL L^−1^ on total soluble solids (TSS), titratable acidity (TA) and ethanol accumulation following storing at 20 °C.

Quality Parameters/Dipping Treatments (1000 μL L^−1^)	Weeks				
	1	2	3	4	Mean
TSS (%)					
Time—0	10.0				
Control	10.1 ± 0.8	10.3 ± 0.7	10.6 ± 0.5	11.0 ± 0.6	10.5 ± 0.8
LM EO	10.4 ± 1.0	10.5 ± 0.9	10.8 ± 0.9	11.0 ± 0.9	10.6 ± 0.9
Citral	10.1 ± 0.9	10.6 ± 0.7	10.8 ± 0.6	11.3 ± 0.7	10.7 ± 0.8
*LSD*					*0.3*
TA (% citric acid)					
Time—0	1.3				
Control	1.2 ± 0.1	1.1 ± 0.2	1.1 ± 0.2	0.9 ± 0.2	1.1 ± 0.2
LM EO	1.2 ± 0.1	1.2 ± 0.1	1.1 ± 0.1	1.0 ± 0.1	1.1 ± 0.1
Citral	1.2 ± 0.1	1.1 ± 0.1	1.0 ± 0.2	0.9 ± 0.1	1.1 ± 0.2
*LSD*					*0.1*
Ethanol accumulation (μL L^−1^)					
Time—0	1.1				
Control	1.3 ± 0.6	1.4 ± 0.5	1.6 ± 1.3	1.7 ± 0.6	1.5 ± 0.8
LM EO	1.2 ± 0.4	1.4 ± 0.6	1.6 ± 0.8	1.7 ± 0.7	1.5 ± 0.6
Citral	1.3 ± 0.6	1.5 ± 0.7	1.7 ± 0.8	1.7 ± 0.7	1.5 ± 0.7
*LSD*					*0.3*
Ethylene production (μLC_2_H_4_ kg^−1^ h^−1^)					
Time—0	1.1 × 10^−5^				
Control	1.1 × 10^−5^	1.4 × 10^−5^	1.3 × 10^−5^	1.4 × 10^−5^	1.3 × 10^−5^
LM EO	1.0 × 10^−5^	1.4 × 10^−5^	1.4 × 10^−5^	1.6 × 10^−5^	1.4 × 10^−5^
Citral	1.1 × 10^−5^	1.4 × 10^−5^	1.4 × 10^−5^	1.6 × 10^−5^	1.4 × 10^−5^
*LSD*					*2.8 × 10^−6^*

Mean of 4 units of 20 fruits ± standard deviation (1 group × 4 repeats). Both the mean value for ethylene production and its standard deviation (SD) were found to be very low, so SD is not provided with the mean.

## Data Availability

All data are presented in the article.

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
