# Peer review of "Efficacy of Lemon Myrtle Essential Oil as a Bio-Fungicide in Inhibiting Citrus Green Mould"

_plants, 2023, doi:10.3390/plants12213742_

Round 1

Reviewer 1 Report (New Reviewer)

Comments and Suggestions for Authors

The work entitled "Efficacy of Lemon Myrtle Essential Oil as a Bio-fungicide in Inhibiting Citrus Green Mould " presents compelling findings. I would like to recommend the manuscript for publication with minor revision. This work provides a comprehensive introduction to the presented problem and presents detailed information. All the necessary data are described, and the main conclusions have been drawn.

Below is a list of comments.

1) An entry of the type "1.3E-05" should be presented as an appropriate power of 10.

2) Figure 1. Fungal wounds in Valencia oranges during storage at 20 °C following dipping in 1000 µL 354 L -1 lemon myrtle essential oil (LM EO) solution for different times. - Because changes over time are shown, the curve should be fitted to an appropriate mathematical model or function.

Author Response

Reviewer 2 Report (New Reviewer)

Comments and Suggestions for Authors

The article written by the authors presents a well-designed investigation into the potential use of essential oil (EO) derived Backhousia citriodora as a natural antifungal agent against Penicillium digitatum, a common post-harvest pathogen affecting citrus fruits.

The authors employed a comprehensive approach, utilizing many laboratory techniques and tests to do a wide-ranging investigation:  a study of the essential oil composition, and its antimicrobial properties, along with an evaluation of its use as a fungicide to store citrus fruits. The authors included required control samples. The research is methodologically sound, providing valuable insights into the potential of studied essential oil as a natural alternative to synthetic fungicides, addressing the issue of wastage in citrus storage. The findings suggest promising applications for studied EO in the agricultural and post-harvest industries, contributing to environmentally friendly practices.

The results demonstrated that LM EO, particularly at higher concentrations (4 and 5 μL disc-1), significantly inhibited fungal growth, similar to the inhibitory effects observed with pure citral. Furthermore, the study evaluated the practical application of LM EO by dipping inoculated oranges in varying concentrations, presenting its potential to reduce fungal wounds without compromising the overall quality and taste of the fruit.

There are some minor errors found and below are suggestions for authors to include, without diminishing the value of their work. These are:

137 – Citrus sinesis should be italicized

245 –  extra space found

Table 1 – 49.5 is moved to the right

Table 1. If possible, please express the retention times as linear retention indices which are more universal than retention times. To validate the accuracy of the MS spectra match, please provide in the table the comparison between the experimental LRI and reference LRI for the identified analytes. You can find them on NIST Chemistry WebBook, SRD 69 website, or in publication  doi:10.1063/1.3653552.

Author Response

Reviewer 3 Report (New Reviewer)

Comments and Suggestions for Authors

1. Line 12-13: “disc-1” should be written as “per disc”.

2. Line 164: “P. digitatum” should be written in italics “P. digitatum”.

3. Table 1: the format in Peak Area (%) should be standardized.

4. The values in Table 2-10 should be written as “means ± standard deviations”, and should use alphabet to indicate whether there is Least Significant Difference between the values.

5. There is discrepancy between the values obtained for LM EO 2000 mL/L and Citral 1000 mL/L in Table 4/Table 5 and the values in Table 6/Table 7, especially Citral. Therefore, the same experiment shall be repeated to confirm the results, in order to confirm the contribution of Citral in LM EO in inhibiting fungal growth and in preventing rind injury. This is to avoid the authors from making wrong conclusion (Line 473-474).

6. High concentration of LM EO showed stronger green mould wound inhibition (Table 4) than lower concentration (Table 6). The high concentration LM EO treatments also exhibited lesser rind injury score (Table 5) compared with low concentration (Table 7). Why did the authors use 1000 mL/L LM EO to conduct Quality assessment?

7. Please describe the equation of the% of fruits with fungal wounds in Figure 1.

8. In Agar Diffusion Assay, only 1, 2, 3, 4 or 5mL essential oil were dropped to the 13mm paper disc. How could the authors assure that the essential oil had diffused and fully covered the size of the inhibition zone? The inhibition zone might also come from the contribution of essential oil vapour volatilized from the paper disc.  

9. The greater antifungal efficacy of essential oil in the vapour assay compared to the agar diffusion assay shall be due to the fact that essential oils compose of non-polar molecules. Diffusion of non-polar molecules through agar are more difficult, whereas vapour has direct contact with the fungal that could inhibit the growth directly. Therefore, the conclusion made in Line 407 “…due to the vapour providing better penetration into the agar media than through the diffusion assay…” is not correct

Round 2

Reviewer 3 Report (New Reviewer)

Comments and Suggestions for Authors

My previous comment (point 5) that there is discrepancy between the values obtained for LM EO 2000 μL/L and Citral 1000 μL/L in Table 4/Table 5 and the values in Table 6/Table 7, especially Citral. While the authors’ answer was: “When we conducted in vivo experiments, we were expecting the trend of the effect/results instead of the values similarity of effects/results.” Since the authors proposed that the experiments were only to show the trend of the effect, the authors should not state in this paper that (line 444-445) "The optimum effectiveness was achieved by 1000 μL L-1 LM EO" whereby 1000 μL L-1 is a questionable/not confirmed value.

Author Response

This manuscript is a resubmission of an earlier submission. The following is a list of the peer review reports and author responses from that submission.

Round 1

Reviewer 1 Report

Comments and Suggestions for Authors

The application of Backhousia citriodora essential oil for the preventive antifungal treatment of citrus fruits is undeniably interesting and attractive.

Unfortunately, the quality of this work is weakened by the fact that the authors tested a commercial essential oil, without any certainty about its quality and authenticity, and an essential oil prepared on a small scale in the laboratory by the non-industrial MAHD technique. In both cases, the representativeness of the essential oil tested for the targeted application is not validated.

While the authors describe in detail in paragraph 2.5 the analytical method (GC-MS) used to control the essential oils, they have omitted to give the results of these analyzes in a table, including the essential data to ensure the identification and quantification of constituents. The values ​​briefly quoted in paragraph 3.1 are not acceptable. This necessarily includes the measurement of retention indices in comparison with the known values ​​of reference compounds, and the quantification at least by internal normalization, from the response of the FID and not that of the TIC of the MS.

The repeated citation of ethylbenzene (lines 247 to 436) as a constituent of the essential oil poses a serious problem, because this hydrocarbon is not a natural product. According to this referee, it has not been cited previously in this essential oil by various authors who have studied it in detail (Southwell et al, Brophy et al. etc.)

More regrettable is the ignorance of the authors of several recent articles on the subject, in particular the review article of Southwell in Foods, 2021, 10, 1596.

We can also cite the articles by A. C. Lim et al. Molecules, 2022, 27, 4895, the chapter by Y.  Sultanbawa, in the book “Essential oils in Food Preservation...” Academic Press (1996).

And other very recent review articles :

D.  Saifullah et al. (2022) Phytochemicals and Bioactivities of Australian Native Lemon Myrtle (Backhousia citriodora) and Lemon-Scented Tea Tree (Leptospermum petersonii): A Comprehensive Review, Food Reviews International, DOI: 10.1080/87559129.2022.2130353

This referee has also noticed that one of the authors  has recently published another article on this subject : M. M. Rahman et al, J. Hort. Sci. Biotechnol., 2022, 97, 524.

This should force authors to drastically shorten the introductory part of their manuscript

Reviewer 2 Report

Comments and Suggestions for Authors

Dear Authors, I reviewed your article. It's well written and presented. I found a few minor edits to the English language in the result and discussion parts. I would like to suggest that you increase the length of the conclusion section. It seems too concise. For the rest of the article, I am in favour of publication. 

Comments on the Quality of English Language

 I found a few minor edits to the English language in the result and discussion parts.

Reviewer 3 Report

Comments and Suggestions for Authors

In Material and methods you wrote various time periods line 143 what you mean by this

What the source of the pathogen and how you identified it? 

How many replicate you used sometime you wrote 3 sometime 6 which one right?

All Tables need to improve all are confused for the reader e.g Table 1 you wrote inhibition % above the concentration

Table 2 you wrote Decay above the days after inoculation

Table 3 the same and you wrote 1.2.3 ….. and all this mentioned in Material and Methods

Table 4 I suggest to change to chart

Table 7 you wrote week what is week?

 English language need to improve

Discussion part need to improve here some new references maybe you can used for improve your discussion. If found useful, cite these recent references

Junior O.J.C.,Youssef K., Koyama R., Ahmed S., Dominguez A.R.,Mühlbeier D.T., Roberto S.R. 2019. Control of Gray Mold on Clamshell-Packaged ‘Benitaka’ Table Grapes Using Sulphur Dioxide Pads and Perforated Liners. Pathogens, 8, 271.https://doi.org/10.3390/pathogens8040271

Hadeel M. M. Khalil Bagy, Badawy F. M. Ibtesam, Eman A. A. Abou-Zaid, Badawy M. Sabah & Sallam, Nashwa, M. A (2021) Control of green mold disease using chitosan and its effect on orange properties during cold storage, Archives of Phytopathology and Plant Protection, 54 (11-12): 570-585   DOI: 10.1080/03235408.2020.1847568

Abo-Elyousr, K.A.M., Al-Qurashi, A.D. & Almasoudi, N.M. Evaluation of the synergy between Schwanniomyces vanrijiae and propolis in the control of Penicillium digitatum on lemons. Egypt J Biol Pest Control 31, 66 (2021). https://doi.org/10.1186/s41938-021-00415-4

Mohamed I. Elsayed, Adel D. Al-Qurashi, Najeeb Marei Almasaudi and, Abo-Elyousr KAM 2022. Efficacy of some essential oils against gray mold of table grapes and their effect on fruit quality. South African Journal of Botany 146 (2022) 481_490 https://doi.org/10.1016/j.sajb.2021.11.046

Hadeel, MM. Kalil Bagy, Abo-Elyousr KAM, Abd El-Latif Hesham, Sallam, Nashwa, M. A. 2023. Development of antagonistic yeasts for controlling black mold disease of Onion. Egyptian Journal Biological Pest Control 33: 17: https://doi.org/10.1186/s41938-023-00664-5

Comments on the Quality of English Language

Moderate editing of English language required

Round 2

Reviewer 1 Report

Comments and Suggestions for Authors

none.
